# The Role of Annexin A1 in DNA Damage Response in Placental Cells: Impact on Gestational Diabetes Mellitus

**DOI:** 10.3390/ijms241210155

**Published:** 2023-06-15

**Authors:** Jusciele Brogin Moreli, Mayk Ricardo dos Santos, Iracema de Mattos Paranhos Calderon, Cristina Bichels Hebeda, Sandra Helena Poliselli Farsky, Estela Bevilacqua, Sonia Maria Oliani

**Affiliations:** 1Post-Graduation in Structural and Functional Biology, Federal University of São Paulo (UNIFESP), São Paulo 04023-062, Brazil; juscielemoreli@gmail.com; 2Faceres School of Medicine (FACERES), São José do Rio Preto 15090-305, Brazil; 3Department of Biology, School of Biosciences, Humanities and Exact Sciences, São Paulo State University (UNESP), São José do Rio Preto 15054-000, Brazil; mayk.kd@gmail.com; 4Graduate Program in Gynecology, Obstetrics and Mastology, Botucatu Medical School, São Paulo State University (UNESP), Botucatu 18618-687, Brazil; iracema.calderon@gmail.com; 5Department of Clinical and Toxicological Analyses, Faculty of Pharmaceutical Sciences, University of Sao Paulo (USP), São Paulo 05508-000, Brazil; crisbh@gmail.com (C.B.H.); sfarsky@usp.br (S.H.P.F.); 6Department of Cell and Developmental Biology, Institute of Biomedical Sciences, University of São Paulo (USP), São Paulo 05508-000, Brazil; bevilacq@usp.br; 7Advanced Research Center in Medicine (CEPAM), União das Faculdades dos Grandes Lagos (Unilago), São José do Rio Preto 15030-070, Brazil

**Keywords:** apoptosis, oxidative stress, base excision repair, annexin A1 knockout mice, high-risk pregnancy

## Abstract

The functions of annexin A1 (ANXA1), which is expressed on membranes and in cytoplasmic granules, have been fully described. Nonetheless, the role of this protein in protecting against DNA damage in the nucleus is still emerging and requires further investigation. Here, we investigated the involvement of ANXA1 in the DNA damage response in placental cells. Placenta was collected from ANXA1 knockout mice (AnxA1^−/−^) and pregnant women with gestational diabetes mellitus (GDM). The placental morphology and ANXA1 expression, which are related to the modulation of cellular response markers in the presence of DNA damage, were analyzed. The total area of AnxA1^−/−^ placenta was smaller due to a reduced labyrinth zone, enhanced DNA damage, and impaired base excision repair (BER) enzymes, which resulted in the induction of apoptosis in the labyrinthine and junctional layers. The placentas of pregnant women with GDM showed reduced expression of AnxA1 in the villous compartment, increased DNA damage, apoptosis, and a reduction of enzymes involved in the BER pathway. Our translational data provide valuable insights into the possible involvement of ANXA1 in the response of placental cells to oxidative DNA damage and represent an advancement in investigations into the mechanisms involved in placental biology.

## 1. Introduction

Annexin A1 (ANXA1) is a member of the annexin superfamily of calcium- and phospholipid-binding proteins [1]. The protein core comprises 346 amino acids (37 kDa) with C- and N-terminal domains, the latter of which confers specificity and physiological activity to each annexin [2]. Besides being characterized as a glucocorticoid-regulated anti-inflammatory protein [3], ANXA1 has also been reported to be involved in critical pathophysiological processes, including cell proliferation, differentiation, and apoptosis of epithelial and cancer cells, which implicates this protein in tissue repair and cancer metastasis [4,5,6].

ANXA1 is found in three distinct subcellular locations: the cytoplasm, nucleus, and plasma membrane. Although none of the annexins contain nuclear-targeting sequences, nuclear localization of ANXA1 has been reported under certain conditions [7]. The translocation of ANXA1 from the cytoplasm to the nucleus is believed to start with mitogenic/proliferative signals or after a DNA-damaging stimulus, such as oxidative stress [7]. In breast cancer cells, nuclear localization of ANXA1 was associated with protection against heat-induced DNA damage [8].

ANXA1 contains a DNA- and/or RNA-binding sequence [9] and has been proposed to perform helicase activity in the nucleus [10,11]. Because helicase activity is required in DNA replication and repair, nuclear ANXA1 may participate in tumorigenesis [6]. DNA lesions, such as double-strand breaks and oxidation of DNA bases, especially guanine, are also induced by hyperglycemia-mediated oxidative stress [12,13,14,15]. We previously demonstrated that maternal hyperglycemia levels are directly proportional to DNA damage and inversely proportional to the expression of base excision repair (BER) enzymes in peripheral blood cells [15]. However, this relationship was not observed in the same cell types in newborns because of the regulation of expression of BER enzymes [15]. BER is the most efficient mechanism for repairing endogenous DNA damage, in which DNA glycosylase (8-oxo guanine DNA glycosylase [OGG1]) removes the damaged base, resulting in an apurinic/apyrimidinic (AP) site. The AP endonuclease (AP endonuclease 1 [APE1]) cleaves the AP site, allowing DNA polymerase to synthesize the repair patch. The latter is re-ligated using the DNA ligase III activity [16].

As a determinant condition of hyperglycemia, GDM, defined as any degree of glucose intolerance with onset or first recognition during pregnancy, is among the most common complications associated with pregnancy [17]. Maternal metabolic changes can, directly and indirectly, be reflected in the placenta, interposed between the maternal and fetal circulation [18], and can lead to impaired embryonic and fetal development [19,20,21]. Cell death [22], inflammation [23,24], and oxidative stress [25] are important hallmarks of placental cell damage in mothers with GDM.

Because nuclear ANXA1 is apparently involved in DNA replication and repair, we tested the hypothesis that nuclear ANXA1 is associated with the placental cellular response induced by oxidative DNA damage. To this end, we used placental samples from AnxA1 knockout animals (AnxA1^−/−^) and from mothers with GDM, a classical condition of oxidative stress and inflammation during pregnancy.

## 2. Results

### 2.1. Deficiency of AnxA1 Gene Modifies the Placental Morphology

The labyrinthine and junctional zones of mouse placenta were identified and measured using histological sections (Figure 1). The mice placentas comprised the following distinct layers: the labyrinth, junctional zone, and maternal decidua (Figure 1A,B). The junctional zone contained spongiotrophoblasts, glycogen, and giant trophoblasts (Figure 1C,D). The labyrinth (Figure 1E,F)—the site of gas and nutrient exchange—was formed by a layer of giant trophoblasts in contact with the maternal blood and syncytiotrophoblast layers facing the fetal capillaries. 

The placentas of AnxA1^−/−^ mice showed a smaller total area (Figure 1G), no difference in junctional zone (Figure 1H) and reduction of labyrinthine zone (Figure 1I) when compared with WT placentas.

### 2.2. Oxidative DNA Damage Is Augmented, and the Expression of Repair Enzymes Is Impaired in Placenta from AnxA1^−/−^ Mice

In placenta from AnxA1^−/−^ mice, the oxidative DNA damage, evaluated based on positive nuclear staining for 8-oxoguanine, was increased in the junctional zone (Figure 2A,B,F) and in the labyrinth (Figure 2C,D,G). The expression of the OGG1 DNA repair enzyme was less in AnxA1^−/−^ in the placental nuclei of both junctional (Figure 2H,I,M) and labyrinthine (Figure 2J,K,N) zones than in the placenta from WT mice. Similarly, the expression of the other component in the DNA repair pathway, namely APE-1, was also reduced in the placenta of AnxA1^−/−^ mice (Figure 2O–U).

### 2.3. DNA Double-Strand Breaks and Apoptosis Are Enhanced in Placenta from AnxA1^−/−^ Mice

The placental cells in the AnxA1^−/−^ group showed increased expression of gamma H2AX in the junctional zone (Figure 3A,B,F) and labyrinth (Figure 3C,D,G) layers. The number of cells showing the expression of cleaved caspase 3 was increased only in the labyrinth (Figure 3J,K,N).

### 2.4. GDM Alters Maternal Clinical Parameters

Pregnant women with GDM presented with higher levels of HbA1c and their newborns were heavier than those of ND women. Other clinical parameters, such as age, BMI, pregnancy weight gain, and placental weight, showed no differences between the GDM and ND pregnant women (Table 1).

### 2.5. GDM Changes the Villous Morphology, Leads to Oxidative DNA Damage, and Impairs the Expression of Nuclear DNA Repair Enzymes

As observed in the placentas from ND women (Figure 4A,C,E), the intermediate villus components were preserved in the placentas from women with GDM (Figure 4B,D,F). Syncytiotrophoblast lined the villi, in contact with the intervillous space (filled with maternal blood), which contained vessels and mesenchyme cells. Nuclear aggregates in syncytiotrophoblast (syncytial knots) and fibrin deposits were observed, especially in the GDM group (Figure 4B; Appendix A).

Oxidative DNA damage was evaluated using nuclear 8-Oxoguanine detection. Cells positive for nuclear staining of 8-Oxoguanine were observed in the syncytiotrophoblast layer, and mesenchymal and endothelial cells in both groups (Figure 5A,B). Quantitative analysis revealed increased DNA damage in placenta samples, particularly in the syncytiotrophoblast from the GDM group (Figure 5B,C).

OGG1 (Figure 5D,E) and APE-1 (Figure 5G,H), the enzymes in the BER pathway, were expressed in syncytiotrophoblast, mesenchymal cells, and endothelial cells in the ND group. Compared with the ND group, lower expression of nuclear BER enzymes (Figure 5F,I) was found in placentas from the GDM group, mainly because of reduced expression in the syncytiotrophoblast (Figure 5E,H).

### 2.6. Placenta from Women with GDM Presents Augmented DNA Double-Strand Breaks and Apoptosis

Double-stranded DNA breaks were evaluated by detecting H2AX phosphorylation at serine 139. Placenta cells exposed to hyperglycemia showed increased DNA double-strand breaks (Figure 6A–C) and apoptotic cells (Figure 6D–F) compared with those in the ND group, as evaluated using activated caspase 3.

### 2.7. ANXA1 Expression Is Impaired in the Placenta from Women with GDM

In both the groups, ANXA1-positive cells were observed to be present in different cellular components of the placental sections (Figure 7). In chorionic villi, ANXA1 was strongly immunoreactive in the cytoplasm and nuclei of syncytiotrophoblast, and weakly stained in mesenchymal cells (Figure 7A–D). In both syncytiotrophoblast and mesenchymal cells, quantitative analysis of the staining in cytoplasm and nuclei indicated reduced ANXA1 expression in the placentas of GDM women (Figure 7E,F).

Immunoreactive bands (~37 kDa) were detected using the ANXA1 antibody in the extracts of villous placental tissue of pregnant women in both the ND and GDM groups. Densitometric analyses of these bands confirmed a reduction in the expression of ANXA1 in placentas from the GDM group (Figure 7G).

## 3. Discussion

ANXA1 is synthesized by immune, epithelial, and cancer cells via the action of different chemical mediators, such as glucocorticoids and cytokines [26,27]. ANXA1, stored in cytoplasmic granules and released into the extracellular compartment, binds to membrane formyl peptides for downstream intracellular signaling of anti-inflammatory, proliferative, apoptosis, and migration processes [26,28,29]. Moreover, ANXA1 is also found in the nucleus, where a protective action against DNA damage has recently been proposed [30]. This study also highlights the nuclear localization of ANXA1. Here, the translational data showed that the reduced expression of ANXA1 in the placenta was associated with increased apoptosis, which may reflect a failure to protect DNA from oxidative damage. In this context, modulation of BER enzymes is a possible underlying mechanism.

Placentas deficient in AnxA1 showed increased oxidative DNA damage, reduced expression of nuclear BER enzymes with DNA double-strand breaks, and apoptosis induction. This study found gH2Ax was detected both in the labyrinth and junctional zone. However, increased apoptosis and structural reduction were only detected in the labyrinth. Although we have not studied the possible bases of this difference, a study using giant trophoblastic cells obtained from trophoblastic stem cells showed that these cells can resist apoptosis induced by DNA damage [31]. This resistance is associated mainly with the selective upregulation of p21. The presence of p21 in the cytoplasm of giant cells prevented apoptosis induced by DNA damage, similar to that found in cancer cells able to escape apoptosis. Giant cells constitute a significant component of the junctional zone, which may be an explanation for why lesser apoptosis and maintenance of the junctional layer were found in our KO animals.

Analysis of placentas exposed to an adverse environment, such as hyperglycemia, showed similar outcomes associated with ANXA1 deficiency in the nucleus and cytoplasm of GDM samples.

It is important to emphasize that pregnant women in the GDM group, although treated with a combination of insulin and diet, showed increased HbA1c levels and weight of newborn compared with those in the ND group. The exaggerated supply of glucose from the mother to the fetus explains the weight of the newborns observed in this study [32]. The degree of maternal glucose tolerance is not reflected in clinical aspects. Changes in placental morphology and physiology have been observed in different degrees of hyperglycemic disorders, including GDM [33]. Previous studies have demonstrated that GDM placentas show a higher expression of inflammasome pathway components [24], with an exacerbation in the production of inflammatory cytokines, oxidative stress, DNA damage response, and apoptotic cells [15,22,24,25]. As these effects were detected in the placenta samples from AnxA1^−/−^ mice, our data strongly suggest a protective role for ANXA1 in placental physiology, which can be mitigated in certain adverse conditions, such as hyperglycemia.

An imbalance in the expression of ANXA1 in the placenta of high-risk pregnancies has been previously described. ANXA1 showed protective activity against *Toxoplasma gondii* infection, based on the observation that the third-trimester placentas expressing lesser ANXA1 were more permissive to this infection than the first-trimester placentas expressing more ANXA1. In the third-trimester placental villous explant culture, parasite reduction was observed after treatment with an ANXA1 peptide mimetic (Ac2-26) [34]. Reduced expression of placental AnxA1 was observed in mothers post ZIKA infection. The placentas of these mothers showed an increased inflammatory response and impaired tissue repair [35]. These data may be related to a deficiency in ANXA1, which consolidates the importance of ANXA1 in the placental response to aggression.

As discussed in a review by Boudhraa [6], ANXA1 can translocate to the nucleus in response to mitogenic/proliferative and DNA-damaging stimuli, such as hydrogen peroxide. In the nuclear compartment, ANXA1 binds to DNA, where helicase activity has been proposed, allowing it to include ANXA1 in DNA synthesis and repair mechanisms [10]. In MCF-7 breast cancer cells, the loss of ANXA1 upon stress led to an increased susceptibility to DNA damage and mutation [8]. However, experiments performed in nasopharyngeal carcinoma cells, suggest that knockdown of ANXA1 inhibits DNA damage by decreasing the generation of intracellular reactive oxygen species and the formation of γ-H2AX and promotes DNA repair by increasing DNA-dependent protein kinase activity [36]. Moreover, prevention of nuclear translocation of ANXA1 using the small peptide Tat-NTS inhibited cellular proliferation (G2/M phase arrest), migration, and invasion of glioblastoma cells [37]. Thus, the DNA damage response related to ANXA1 is tissue-dependent and the helicase activity proposed to ANXA1 may explain the involvement of this protein with DNA repair mechanisms.

Our data indicate crucial functions of ANXA1 in placental cells, especially in the nuclear compartment. Under hyperglycemic conditions (a classical oxidative stress condition), we observed a reduction in nuclear expression of ANXA1 in GDM placentas. As observed in MCF7 breast adenocarcinoma cells, our data indicate that the absence or reduction of ANXA1 in placental cells might underlie the cell death response and impair placental function.

Overall, our translational data indicate that differential expression of ANXA1 in the placenta alters the cellular response to oxidative DNA damage, leading to apoptosis. We focused on the reduction in BER enzymes in the nuclear compartment to explain apoptotic signaling. These findings demonstrate the relevance of ANXA1 in placental cell responses and shed light on the mechanisms that regulate the functions of this critical protein in placental biology.

## 4. Materials and Methods

### 4.1. Ethical Statement

This cross-sectional study included placental samples from AnxA1 knockout (AnxA1^−^/^−^) mice and mothers with GDM. All animal procedures were performed according to the Brazilian Society for the Science of Laboratory Animals (SBCAL) and were approved by the Institutional Animal Care and Use Committee of the Faculty of Pharmaceutical Sciences at the University of Sao Paulo (Protocol 521). Placental samples from pregnant women were obtained from the Diabetes and Pregnancy Service of Botucatu Medical School/UNESP, Brazil, with the approval of the Research Ethics Committee (protocol # 48609715.0.0000.5505). Written informed consent was obtained from all participants according to the principles of the Declaration of Helsinki.

### 4.2. AnxA1 Knockout (AnxA1^−/−^) Placental Samples

Male and female wild type (WT) and AnxA1^−/−^ BALB/c mice, aged 5–6 weeks, were maintained and reproduced at the animal house of the Faculty of Pharmaceutical Sciences, University of Sao Paulo (Brazil). The deficiency of AnxA1 knockout in placentas was demonstrated by immunohistochemical analysis and showed in Appendix A. Animals were provided chow (Nuvilab) and water ad libitum. All animals were housed in a temperature-controlled room (22–25 °C and 70% relative humidity) with a 12 h light-dark cycle. Female mice were caged overnight with males (3:1) and successful mating was verified the following morning by the presence of a vaginal plug (day 0.5 of pregnancy). On day 18.5 of pregnancy, mice were euthanized by cervical dislocation or were anesthetized with xylazine and ketamine (i.p., 7 and 77 mg/kg, respectively; Vetbrands, Jacarei, SP, Brazil) [38] for collection of placenta samples (six from each group: WT and AnxA1^−/−^). The placenta samples (n = 6/group) were processed for morphological (hematoxylin-eosin staining) and immunohistochemical analyses. 

### 4.3. Population Characterization and Collection of Human Placenta Samples

GDM was diagnosed using a 75 g glucose tolerance test (75 g-GTT), as recommended by the American Diabetes Association [17], between the 24th and 28th gestational weeks. Twenty placenta samples were used in this study, ten each from the nondiabetic (ND; normal 75 g-GTT) and GDM (abnormal 75 g-GTT) groups.

Population characteristics included age, body mass index (BMI) in the third trimester of pregnancy, weight gain during pregnancy, gestational age at delivery, glycemic mean (GM), and glycated hemoglobin (HbA1c) levels. GM was calculated from the arithmetic mean of plasma glucose levels measured in all glucose profiles obtained during treatment (diet or diet + insulin). Plasma glucose levels were measured using the oxidase method (Glucose Analyzer II Beckman, Fullerton, CA, USA) and HbA1c levels were measured using high-performance liquid chromatography (D10™ Hemoglobin Testing System, Bio-Rad Laboratories, Hercules, CA, USA). Placental and fetal weights were also included.

The inclusion criteria were as follows: (i) hyperglycemia defined at a minimum gestational age of 28–30 weeks; (ii) prenatal and delivery care at the service; (iii) absence of clinically diagnosed infections and negative serology for HIV and syphilis, absence of multiple pregnancies, overt diabetes, fetal malformations, fetal death, alcohol consumption, or illicit drug habits; and (iv) deliveries before the 36th week of gestation.

Placentas were collected immediately after delivery (cesarean section) and cut into smaller fragments from different cotyledons. Decidual and villous areas were dissected and rinsed in phosphate-buffered saline. A part of the placental villous area was subjected to routine morphological and immunohistochemical procedures, and the remaining part was frozen and stored at −80 °C for western blotting.

### 4.4. Preparation of Placenta Samples for Histological Assessment

Placenta samples from pregnant women with GDM and from AnxA1^−/−^ mice were fixed in 4% buffered paraformaldehyde for 24 h, dehydrated in a graded ethanol series, and embedded in paraffin (Merck, Darmstadt, Germany). Three-micrometer thick sections were obtained for hematoxylin-eosin staining and immunohistochemistry.

### 4.5. Morphological and Morphometric Analyses of Placenta Samples from AnxA1^−/−^ Mice

Morphometric analysis was performed on formalin-fixed placental samples stained with hematoxylin and eosin and scanned using the Image-Pro Plus version 4.5 for Windows software—Zeiss-Jenaval (Zeiss-Jenaval, Jena, Germany). Placental slides from six dams in each group were analyzed using the ImageJ software (National Institutes of Health, Bethesda, MD, USA). The thickest point of the labyrinth or junctional zone was first identified to measure the thickness of the placental layer. The thickness of each layer was measured and the ratio of the thickness of each layer to the total thickness of the placenta was calculated.

### 4.6. Immunohistochemical Detection of Oxidative DNA Damage, DNA-Double Strand Breaks, DNA Repair Enzymes, and Apoptosis in GDM and AnxA1^−/−^ Placenta Samples

The immunohistochemical detection of 8-Oxoguanine (oxidative DNA damage marker), gamma H2AX (DNA-double strand breaks), APE-1 and OGG1 (DNA repair enzymes), and cleaved caspase-3 (apoptosis) in placental sections from GDM and AnxA1^−/−^ mice was performed. Endogenous peroxide activity was blocked using 3% hydrogen peroxide for 30 min. The tissue sections were then incubated with the following primary antibodies overnight at 4 °C: 8-Oxoguanine (Abcam, Cambridge, UK), gamma H2AX (Novus Biologicals, Littleton, CO, USA), OGG1 (Novus Biologicals), APE-1 (Novus Biologicals), polyclonal rabbit anti-caspase-3 (Abcam). For negative control, sections were incubated with 10% TBS-BSA (Sigma–Aldrich, St. Louis, MO, USA) instead of the primary antibody. After washing with TBS, the sections were incubated with horseradish peroxidase-conjugated secondary antibodies (Abcam). Staining was visualized using a 3,3′-diaminobenzidine substrate (Invitrogen, Waltham, MA, USA). Finally, the sections were counterstained with hematoxylin (Inlab Confiança, São Paulo, Brazil) and mounted using Entellan (Merck, Germany).

### 4.7. Nuclear OGG1, APE-1, and Caspase-3 Analysis in Placenta from AnxA1^−/−^ Mice

Cells with positive nuclear staining for OGG1 and APE-1 in the immunohistochemical analysis were counted in ten fields (×40) in each placental zone (labyrinthine or junctional) for all placenta samples. The ImageJ software (NIH) was used to quantify the area and number of positive nuclei to calculate the ratio of positive nuclei per placental zone.

Data are presented as the number of positive cells per 10^4^ µm^2^ area. Cleaved caspase-3 analysis was performed similarly, considering the number of positive cells. Images were obtained using an Axioskop2-Mot Plus Microscope (Carl Zeiss, Jena, Germany) with the AxioVision software (SE64 Rel. 4.9.1).

### 4.8. Morphological Analysis of Placenta from Pregnant Women with GDM

Morphological analysis was performed on formalin-fixed placenta samples stained with hematoxylin-eosin and cytokeratin 7. The expression of cytokeratin 7 (a specific marker for trophoblast cells) was detected immunohistochemically using polyclonal rabbit IgG anti-cytokeratin 7 (Abcam; 1:50). Images were obtained using an Axioskop 2-Mot Plus Microscope (Carl Zeiss) with the AxioVision software.

### 4.9. Analysis of Nuclear OGG1, APE-1, and Caspase-3 in Placenta from Pregnant Women with GDM

Cells with positive nuclear staining for OGG1 and APE-1 in the villous area were counted in ten fields (×40) of each placenta. As described for mouse placentas, the ImageJ software (NIH) was used to quantify the villous area, and the ratio between the number of positive nuclei and villous area was calculated. Data are presented as the number of positive cells per 10^4^ µm^2^ area. Cleaved caspase-3 analysis was performed considering positive cells.

### 4.10. Detection and Analysis of Cytoplasmic and Nuclear ANXA1 in GDM Placentas

The expression of ANXA1 was determined by immunohistochemical staining as described above, using the following primary antibody: polyclonal rabbit anti-AnxA1 (Zymed Laboratories, Cambridge, UK) at 1:5000 for 1 h.

The intensity of cytoplasmic ANXA1 staining in villous cells was measured densitometrically using 10^3^ random points from ten fields (×40) from each placenta on an arbitrary scale from 0 to 255.

Cells with positive nuclear staining for ANXA1 in the villous area were counted in ten fields (×40) of each placenta. The ImageJ software (NIH) was used to quantify the villous area and the ratio of positive nuclei to the villous area was calculated. Data are presented as the number of positive cells per 10^4^ µm^2^ area. Images were obtained using an Axioskop 2-Mot Plus Microscope (Carl Zeiss, Jena, Germany) with the AxioVision software (SE64 Rel. 4.9.1).

### 4.11. Western Blot Analysis of ANXA1 Using Placental Extracts from Patients with GDM

Villous fragments of frozen human placental tissues were transferred to propylene tubes containing the lysis buffer (Merck, Darmstadt, Germany) and a cocktail of protease inhibitors (Complete Mini, EDTA-free protease inhibitor cocktail tablets, Roche, Switzerland). Samples were homogenized on ice using an electric homogenizer. The homogenates were centrifuged at 12,000 rpm for 15 min at 4 °C. The protein concentration in the supernatant was measured using a BCA protein assay (Pierce^TM^ BCA Protein Assay Kit, Thermo Scientific, Waltham, MA, USA) and then stored at −80 °C for western blot analysis.

Equal amounts of total protein from each group were separated electrophoretically on a 15% SDS-polyacrylamide gel and then transferred to a 0.45 μm nitrocellulose membrane (Millipore, Burlington, MA, USA). The transfer of proteins was confirmed by staining the membranes with 10% Ponceau S solution (Sigma Aldrich, St. Louis, MO, USA). The blotted membranes were blocked with 3% TBS-T-milk for 1 h, washed three times with TBS buffer, and then incubated overnight at 4 °C with anti-β-actin (1:2000; Novus Biologicals, Centennial, ON, USA) and anti-ANXA1 (1:5000; Invitrogen, Carlsbad, CA, USA) antibodies in 3% TBS-T-milk and washed three times with TBS buffer. The membranes were exposed to a horseradish peroxidase-conjugated secondary antibody (1:1000; Abcam) in 3% TBS-T-milk for 1 h and washed three times with TBS buffer. Immunoreactive bands (indicative of peroxidase activity) were detected using the enhanced chemiluminescence method. Quantitative analysis of ANXA1 was performed by densitometry using the ImageJ software (NIH, Bethesda, MD, USA). β-Actin was used as a loading control.

### 4.12. Statistical Analysis

Data are presented as mean ± standard deviation (SD). Statistical analyses were performed using the GraphPad software version 6.00. First, we performed the Kolmogorov–Smirnov normality test to determine whether the data distribution was parametric or nonparametric. Student’s *t*-test or the Mann–Whitney *U* test was used for group comparisons. Statistical significance was set at *p* < 0.05.

## Figures and Tables

**Figure 1 ijms-24-10155-f001:**
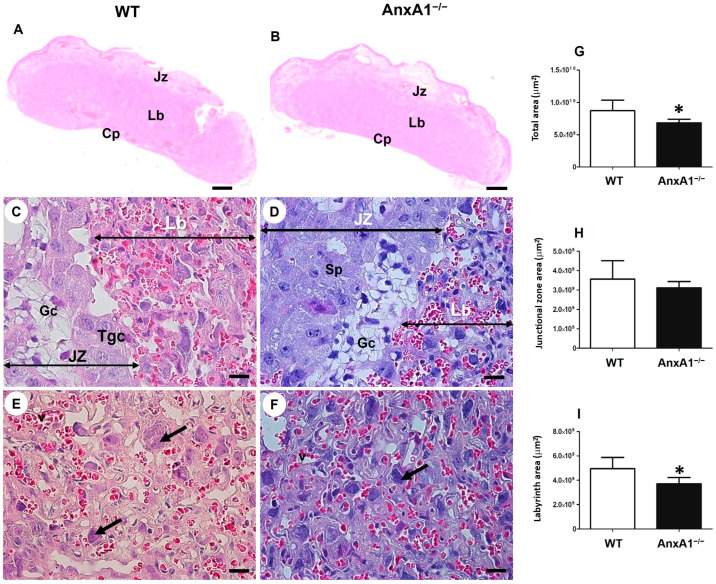
Representative images of placental sections from wild-type (WT) (**A**) and AnxA1 knockout animals (AnxA1^−/−^, (**B**)). The histologic sections show the junctional (Jz) (**C**,**D**) and labyrinth (Lb) (**E**,**F**) placental zones. Scale bars = 20 μm. Jz: junctional zone; Lb: labyrinth; Cp: chorionic plate; Gc: glycogen cells; Sp: spongiotrophoblast; Tgc: junctional zone giant trophoblast cells; arrows: labyrinthine giant trophoblast cells. The sections were stained with hematoxylin and eosin Morphometric analyses of total area (**G**), junctional zone (**H**) and labyrinth (**I**) of placentas from WT and AnxA1^−/−^ animals. Values are shown as mean ± SD. * *p* < 0.05, n = 6/group.

**Figure 2 ijms-24-10155-f002:**
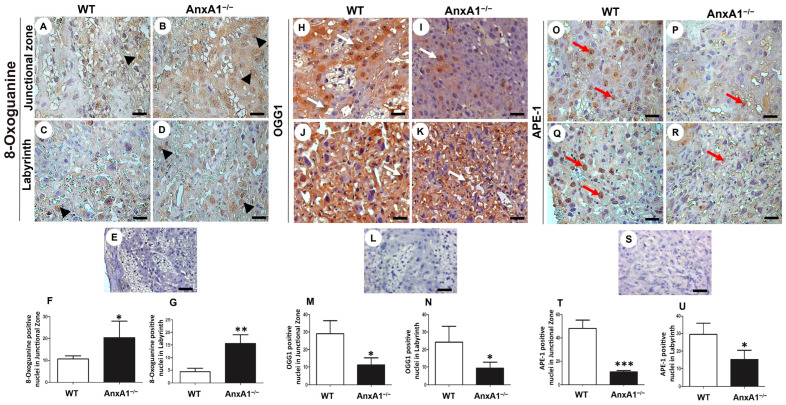
Immunolocalization of 8-Oxoguanine (oxidative DNA damage, (**A**–**G**)) and OGG1 (**H**–**N**) and APE-1 (DNA repair enzymes, (**O**–**U**)) in placenta sections from WT and AnxA1^−/−^ animals. Placental cells reactive to 8-Oxoguanine (arrowheads; (**A**–**D**)), OGG-1 (arrows; (**H**–**K**)), and APE-1 (red arrows; (**O**–**R**)) are found in both junctional and labyrinthine placental zones with a similar pattern of immunoreactivity. (**E**,**L**,**S**) Buffer was used instead of the primary antibody as a negative control, in panels. Bars = 50 μm. (**A**–**E**,**H**–**L**,**O**–**S**) Immunoperoxidase and hematoxylin counterstaining. (**F**,**G**,**M**,**N**,**T**,**U**) Quantification of 8 hydroxyguanosine- (**F**,**G**), OGG-1- (**M**,**N**), and APE-1- (**T**,**U**) positive nuclei per 10,000 μm^2^ of placental area. Values are shown as mean ± SD; * *p* < 0.05; ** *p* < 0.01, *** *p* < 0.001, n = 6/group.

**Figure 3 ijms-24-10155-f003:**
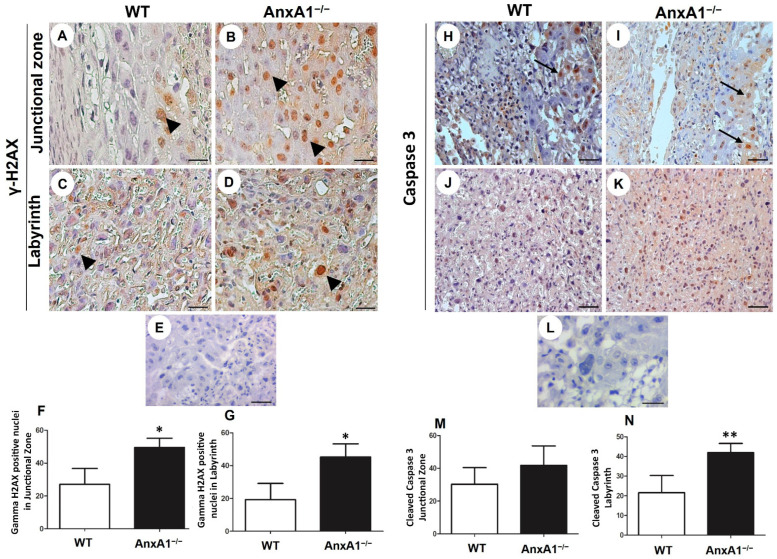
Immunolocalization of γ-H2AX (DNA double-strand breaks, (**A**–**G**)) and cleaved caspase-3 (apoptosis, (**H**–**N**)) in placenta sections from WT and AnxA1^−/−^ animals. (**A**–**E**,**H**–**L**) Placental cells reactive to γ-H2AX (arrowheads; (**A**–**E**)) and caspase-3 (arrows; (**H**–**L**)) are found in both junctional and labyrinthine placental zones. (**E**,**L**) Negative control for immunohistochemical analysis. (**A**–**E**,**H**–**L**) Immunoperoxidase and hematoxylin counterstaining. Bars = 50 μm. (**F**,**G**,**M**,**N**) Quantification of γ-H2AX-stained nuclei (**F**,**G**) and caspase-3-positive cells (**M**,**N**) per 10,000 μm^2^ of placental area. Values are shown as mean ± SD; * *p* < 0.05; ** *p* < 0.01, n = 6/group.

**Figure 4 ijms-24-10155-f004:**
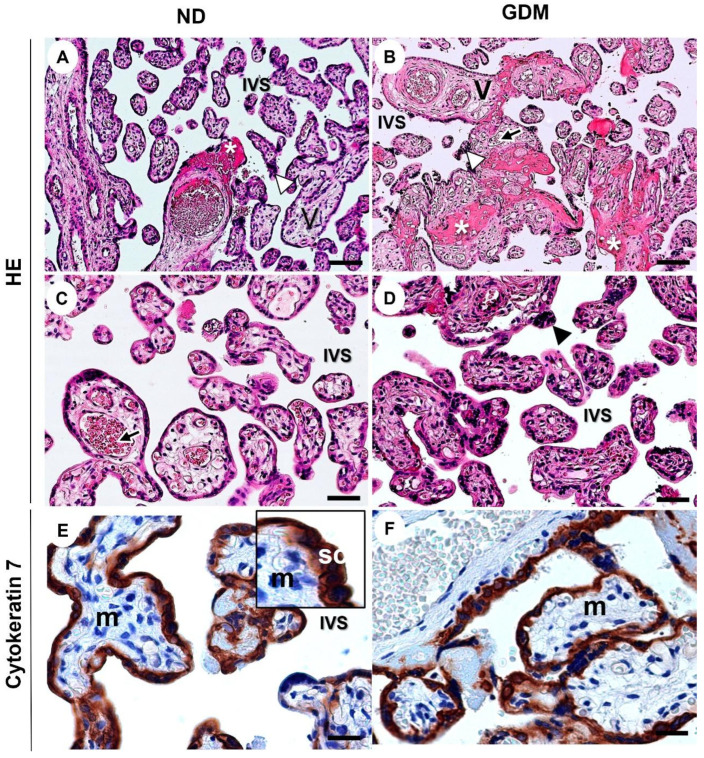
Representative images of villous compartment of term placentas stained with hematoxylin and eosin (**A**–**D**) and immunolabeling of cytokeratin 7 for identification of the syncytiotrophoblast layer (SC) around the mesenchyme (m) from pregnant women with no diabetes (ND; (**A**,**C**,**E**)) or gestational diabetes mellitus (GDM; (**B**,**D**,**F**)). (**A**–**C**) Note the aggregations of syncytiotrophoblast nuclei (arrowhead) and increased intramural fibrinoid (asterisk) mainly in GDM villi (**B**). V: chorionic villous; black arrows: fetal vessels; IVS: intervillous space. Bars = 50 μm.

**Figure 5 ijms-24-10155-f005:**
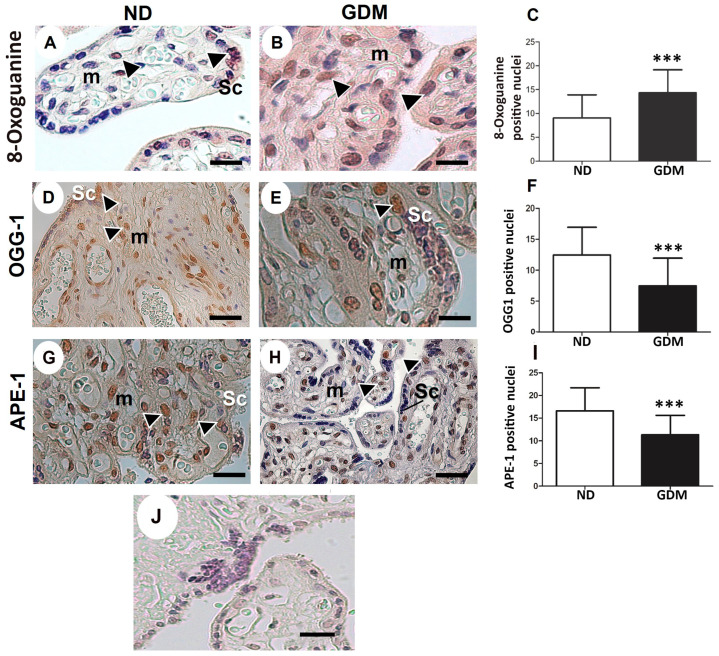
Immunolocalization of oxidative DNA damage (**A**–**C**) and DNA repair enzymes (**D**–**I**) in placentas from pregnant women with no diabetes (ND; (**A**,**D**,**G**)) or gestational diabetes mellitus (GDM; (**B**,**E**,**H**)). Nuclei of the syncytiotrophoblast (arrowheads) and mesenchyme cells (m) were reactive for 8-Oxoguanine (**A**,**B**), OGG-1 (**D**,**E**) and APE-1 (**G**,**H**) in ND and GDM placentas. (**J**) Buffer was used instead of the primary antibody as a negative control. Immunoperoxidase and hematoxylin counterstaining. Bars = 50 μm. (**C**,**F**,**I**) Quantification of 8-Oxoguanine (**C**), OGG-1 (**F**), and APE-1 (**I**) reactive nuclei per 10,000 μm^2^ of placental area. Values are shown as mean ± SD; *** *p* < 0.001, n = 10/group.

**Figure 6 ijms-24-10155-f006:**
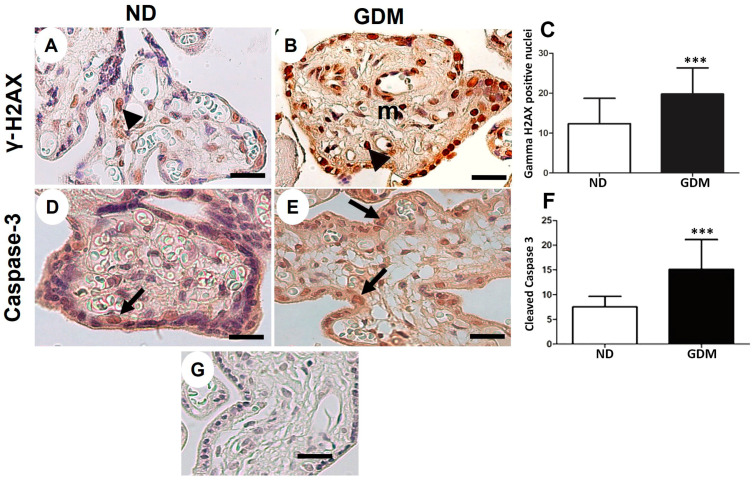
Immunolocalization of γ-H2AX (DNA double-strand breaks, (**A**–**C**)) and cleaved caspase-3 (apoptosis, (**D**–**F**)) in placenta from pregnant women with no diabetes (ND; (**A**,**D**)) or gestational diabetes mellitus (GDM; (**B**,**E**)). (**A**,**B**,**D**,**E**) Nuclei of the syncytiotrophoblast (arrows) and mesenchymal cells (m) are reactive to γ-H2AX (arrowheads). Caspase-3 reactive cells follow a similar pattern of γ-H2AX immunoreactivity (arrows). (**G**) Negative control used in immunohistochemical analysis. Immunoperoxidase and hematoxylin counterstaining. Bars = 50 μm. (**C**,**F**) Quantification of γ-H2AX-positive nuclei (**C**) and caspase-3-reactive cells (**F**) per 10,000 μm^2^ of placental area. Values are shown as mean ± SD; *** *p* < 0.001, n = 10/group.

**Figure 7 ijms-24-10155-f007:**
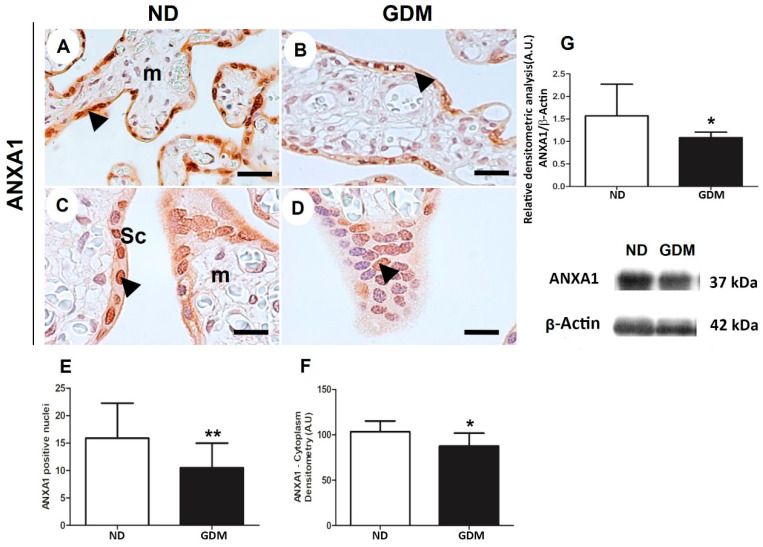
ANXA1 expression in placenta from pregnant women with no diabetes (ND; (**A**,**C**,**G**)) or gestational diabetes mellitus (GDM; (**B**,**D**,**G**)) evaluated by immunohistochemistry (**A**–**F**) and western blot (**G**). Placental reactive components (arrowheads) are mainly nuclei and cytoplasm of syncytiotrophoblast (Sc) and some cells found in mesenchyme (m). The sample were counter-stained with hematoxylin. Bars = 50 μm. 40× (**A**,**B**); 100× (**C**,**D**) Quantification of positive nuclei in 10,000 μ^2^ (**E**) and cytoplasm densitometry (**F**). The relative band intensities from western blot experiments were normalized to β-actin and analyzed with ImageJ software (**G**). Values as mean ± SD; * *p* < 0.05; ** *p* < 0.01, n = 10/group.

**Table 1 ijms-24-10155-t001:** Clinical data.

	ND (n = 10)	GDM (n = 10)
Maternal age (years)	30.85 ± 6.11	29.43 ± 5.36
BMI (Kg/m^2^)	34.88 ± 8.44	38.05 ± 4.37
Pregnancy weight gain (Kg)	12.83 ± 7.50	12.34 ± 6.94
HbA1c (%)	5.36 ± 0.63	6.32 ± 0.82 *
Placental weight (g)	623.10 ± 78.38	632.50 ± 179.80
Newborn weight (g)	3062.00 ± 276.60	3540.00 ± 574.50 *

Data presented as means ± standard deviation. BMI: body mass index evaluated in the third trimester of pregnancy; HbA1c: Glycated hemoglobin evaluated in the third trimester of pregnancy; * (*p* < 0.05).

## Data Availability

The data presented in this study are available on request from the corresponding author.

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
