# Peer review of "The Role of Annexin A1 in DNA Damage Response in Placental Cells: Impact on Gestational Diabetes Mellitus"

_ijms, 2023, doi:10.3390/ijms241210155_

Round 1
Reviewer 1 Report
This manuscript describes role of ANXA1 in protecting placental cells from oxidative DNA damage. Some experiments and clarifications can improve the manuscript. Please find comments below:
Comments:
1. Typo: In Result section “2.1. Deficiency of AnxA1 gene modifies the placental morphology” 2nd line “AnxA1-/- placentas of WT mice” should be “placentas of WT mice”.
2. Authors used 8 hydroxyguanosine as oxidative DNA damage marker (Figures 2 and 5) which is normally used as RNA oxidative damage marker. They should redo the experiment using 8-Oxo-2'-deoxyguanosine (8-oxo-dG) which is more commonly used as oxidative DNA damage marker.
3. As authors find increased positive nuclear staining for 8 hydroxyguanosine in placenta from AnxA1−/− mice’s and in pregnant women with GDM, it’s possible that Annexin A1 might have some roles in protecting nuclear RNA from oxidative damage. Authors should clarify and discuss this in their manuscript.
4. Why does only labyrinthine zone in AnxA1−/− mice’s placenta shows reduction in size and increased apoptosis whereas oxidative damage, gH2Ax were detected both in junctional zone and labyrinthine zone? Authors should clarify and discuss this in their manuscript.
5. How does ANXA1 modulate expression of BER enzymes in placental cells? Authors should clarify and discuss this in their manuscript.
6. Typo: In Result section: “Here, the translational data showed that the expression of ANXA1 in the placenta was associated with increased apoptosis, which may reflect a failure to protect DNA from oxidative damage” should be “Here, the translational data showed that the reduced expression of ANXA1 in the placenta was associated with increased apoptosis, which may reflect a failure to protect DNA from oxidative damage.”
Minor editing required. Please correct the typographic errors
Author Response
June, 05, 2023
Ijms-2418446: The Role of Annexin A1 in DNA Damage Response in Placental Cells: Impact on Gestational Diabetes Mellitus
Caitlin Huang, Ph. D.
Editor-in-Chief
International Journal of Molecular Sciences
Dear Dr Huang,
We are very grateful for the new opportunity to resubmit our article. Changes in the revised manuscript are shown highlighted and a point-to-point response to the Editor comments are also included. We appreciate the reviewer’s constructive criticisms and hope we have been able to address all the concerns.
Best regards,
Sonia Oliani
Revisor 1
Comments and Suggestions for Authors
This manuscript describes the role of ANXA1 in protecting placental cells from oxidative DNA damage. Some experiments and clarifications can improve the manuscript. Please find comments below:
Comments:
- Typo: In Result section “2.1. Deficiency of AnxA1 gene modifies the placental morphology” 2ndline “AnxA1-/- placentas of WT mice” should be “placentas of WT mice”.
Done
- Authors used 8 hydroxyguanosine as oxidative DNA damage marker (Figures 2 and 5) which is normally used as RNA oxidative damage marker. They should redo the experiment using 8-Oxo-2'-deoxyguanosine (8-oxo-dG) which is more commonly used as oxidative DNA damage marker.
- As authors find increased positive nuclear staining for 8 hydroxyguanosine in placenta from AnxA1−/− mice’s and in pregnant women with GDM, it’s possible that Annexin A1 might have some roles in protecting nuclear RNA from oxidative damage. Authors should clarify and discuss this in their manuscript.
We thank the reviewer for highlighting these points in questions 2 and 3. By carefully examining the antibody datasheet (Anti-Oxoguanine 8 antibody [2Q2311] (ABCAM, ab206461)) used in our experiments, it was possible to confirm that the Abcam antibody detects Oxoguanine 8 (8-Oxoguanine), as a mutagenic oxidative damage product of guanine. We apologize for the mistake in the antibody target and changed 8 hydroxyguanosine to 8-Oxoguanine (information about the antibody was obtained from https://www.abcam.com/products/primary-antibodies/oxoguanine-8-antibody-2q2311-ab206461.html). In this context, the discussion was kept focused on guanine as the primary target for reactive oxygen species in DNA, with 8-oxoguanine being the most frequent base lesion and a critical biomarker of oxidative DNA damage.
- Why does only labyrinthine zone in AnxA1−/− mice’s placenta show reduction in size and increased apoptosis whereas oxidative damage, gH2Ax were detected both in junctional zone and labyrinthine zone? Authors should clarify and discuss this in their manuscript.
As suggested, this point has been added to the discussion.
“This study found gH2Ax was detected both in the labyrinth and junctional zone. However, increased apoptosis and structural reduction were only detected in the labyrinth. Although we have not studied the possible bases of this difference, a study using giant trophoblastic cells obtained from trophoblastic stem cells showed that these cells can resist apoptosis induced by DNA damage (de Renty et al., 2014: de Renty C, DePamphilis ML, Ullah Z (2014) Cytoplasmic Localization of p21 Protects Trophoblast Giant Cells from DNA Damage Induced Apoptosis. PLoS ONE 9(5): e97434. https://doi.org/10.1371/journal.pone.0097434). This resistance is associated mainly with the selective upregulation of p21. The presence of p21 in the cytoplasm of giant cells prevented apoptosis induced by DNA damage similar to that found in cancer cells able to escape apoptosis. Giant cells constitute a significant component of the junctional zone, which may be an explanation for why lesser apoptosis and maintenance of the junctional layer were found in our KO animals.’
- How does ANXA1 modulate expression of BER enzymes in placental cells? Authors should clarify and discuss this in their manuscript.
Our translational data indicate that differential expression of ANXA1 in the placenta alters the cellular response to oxidative DNA damage, leading to apoptosis. We focused on reducing BER enzymes in the nuclear compartment to explain apoptotic signaling. Our work did not explore how the ANXA1 modulates BER enzyme expression. However, the literature insights some points that may be considered as the ANXA1 helicase activity that allows it to include ANXA1 in DNA synthesis and repair mechanisms. This information was emphasized in the discussion.
- Typo: In Result section: “Here, the translational data showed that the expression of ANXA1 in the placenta was associated with increased apoptosis, which may reflect a failure to protect DNA from oxidative damage” should be “Here, the translational data showed that the expression of ANXA1 in the placenta was associated with increased apoptosis, which may reflect a failure to protect DNA from oxidative damage”
Done.
Comments on the Quality of English Language
Minor editing required. Please correct the typographic errors
The writing has been reviewed by experts in corrections of scientific articles.

Reviewer 2 Report
In this study, the authors tested the hypothesis that nuclear ANXA1 was associated with the placental cellular response induced by oxidative DNA damage using the ANXA1 knockout mouse model (AnxA1-/-) and pregnant women with gestational diabetes mellitus (GDM). The authors concluded that differential expression of ANXA1 in the placenta alters the cellular response to oxidative DNA damage, leading to apoptosis; the relevance of ANXA1 in placental cell responses and shed light on the mechanisms that regulate the functions of this critical protein in placental biology.
Comments
The reviewer has some concerns as follows:
1. In AnxA1-/- mouse model, the data for ANXA1 gene or protein expression in the placenta should be shown to identify that the deficiency of AnxA1 gene in the placenta is successful. Moreover, the basic information for AnxA1-/- mice should also be shown, such as pregnancy weight gain, placental weight, newborn weight, and blood glucose. The sample size (n number) should also be shown.
2. In Figure 1A and B, one Lb label is near the middle (A), and the other Lb label is near the end (B), is it different?
3. Are there substantial differences in placental histology between the WT and AnxA1 knockout groups? The histological examination should be analyzed and scored.
4. In Table 1, there is no significant difference for the data of placental weight between ND and GDM with 623.10 ± 78.38 vs 179.80 ± 63.5, respectively, please confirm these data.
5. In Figure 4, are there any histopathological changes between ND and GDM groups? The histological examination should be analyzed and scored.
6. The image sizes in Figure 5K is much smaller than others, but the scale bar is the same with others. The scales for the images in Figure 5 need to be clarified. Moreover, the scale bar in Figure 6G is lacking and needs to be added. The sample size (n number) for Figures 5-7 needs to be shown.
7. In Figure 7G, the Western blot analysis is not convincing and needs to be confirmed. In Figure 7, overall, the reduction in protein ANXA1 expression in the placenta from GDM group was so insignificant (only a 1.5-fold difference) compare to the ND group, it was hard to explain its role in the placenta of GDM patients.
8. In general, the presented data cannot support the conclusions.
Author Response
Comments and Suggestions for Authors
In this study, the authors tested the hypothesis that nuclear ANXA1 was associated with the placental cellular response induced by oxidative DNA damage using the ANXA1 knockout mouse model (AnxA1-/-) and pregnant women with gestational diabetes mellitus (GDM). The authors concluded that differential expression of ANXA1 in the placenta alters the cellular response to oxidative DNA damage, leading to apoptosis; the relevance of ANXA1 in placental cell responses and shed light on the mechanisms that regulate the functions of this critical protein in placental biology.
Comments
The reviewer has some concerns as follows:
- In AnxA1-/-mouse model, the data for ANXA1 gene or protein expression in the placenta should be shown to identify that the deficiency of AnxA1 gene in the placenta is successful.
As recommended, the protein expression data was included in the ms. ANXA1 expression was determined by immunohistochemistry in the placentas of WT and AnxA1 -/- groups. These results are summarized in Supplemental Figure 1. Samples size was added in the 4.2 section of materials and methods and figures legends.
Moreover, the basic information (fetal weight, placental weight and placental index) were similar in both groups (as demonstrated below). If necessary, we can add in ms.
Placental weight (g) : WT: 0.67 ± 0.14 ; AnxA1 -/- : 0.85 ± 0.38
Fetal weight (g): WT: 0.13 ± 0.02 ; AnxA1 -/- : 0.18 ± 0.05
Placental index: WT: 5.24 ± 1.29 ; AnxA1 -/- : 5.16 ± 2.91
- In Figure 1A and B, one Lb label is near the middle (A), and the other Lb label is near the end (B), is it different?
As suggested, the layers' identification was standardized in Figures A and B.
- Are there substantial differences in placental histology between the WT and AnxA1 knockout groups? The histological examination should be analyzed and scored.
We analyzed all placental samples and found a reduction in the total and the labyrinth areas in the AnxA1−/− group (Figure 1). No more substantial differences in placental histology were observed.
- In Table 1, there is no significant difference for the data of placental weight between ND and GDM with 623.10 ± 78.38 vs 179.80 ± 63.5, respectively, please confirm these data.
Sorry for the mistake. The correct values are 623.10±78.38 / 632.5±179.80 in ND and GDM, respectively. Table 1 data was revised, and the placental weight was corrected.
- In Figure 4, are there any histopathological changes between ND and GDM groups? The histological examination should be analyzed and scored.
The histopathological analysis was performed using a score adapted from Meyerholz & Beck (2018). Fibrin deposition and syncytiotrophoblast knots were the alterations observed and scored. The results are compiled in Supplementary Table 2.
- The image sizes in Figure 5K are much smaller than others, but the scale bar is the same with others. The scales for the images in Figure 5 need to be clarified. Moreover, the scale bar in Figure 6G is lacking and needs to be added. The sample size (n number) for Figures 5-7 needs to be shown.
The samples size was add in figures legends. The figures bars were revised and the information add in ms.
- In Figure 7G, the Western blot analysis is not convincing and needs to be confirmed. In Figure 7, overall, the reduction in protein ANXA1 expression in the placenta from GDM group was so insignificant (only a 1.5-fold difference) compared to the ND group, it was hard to explain its role in the placenta of GDM patients.
The differences found in WB were indeed slight but still significant. Furthermore, this result was confirmed by immunohistochemical detection of ANXA1 followed by densitometric analysis of this protein in the cytoplasm and the number of positive nuclei. All studies were consistent and ensured the ANXA1 reduction in GDM placentas.

Round 2
Reviewer 2 Report
This revised manuscript can be accepted. No further comments.